# Sturgeon Chondroitin Sulfate Restores the Balance of Gut Microbiota in Colorectal Cancer Bearing Mice

**DOI:** 10.3390/ijms23073723

**Published:** 2022-03-28

**Authors:** Ruiyun Wu, Qian Shen, Pinglan Li, Nan Shang

**Affiliations:** 1Key Laboratory of Precision Nutrition and Food Quality, College of Food Science and Nutritional Engineering, China Agricultural University, Beijing 100083, China; wry0814@cau.edu.cn; 2College of Engineering, China Agricultural University, Beijing 100083, China; 3Department of Biology, Rhodes College, 2000 North Pkwy, Memphis, TN 38112, USA; shenq@rhodes.edu; 4Key Laboratory of Precision Nutrition and Food Quality, Department of Nutrition and Health, China Agricultural University, Beijing 100083, China

**Keywords:** sturgeon chondroitin sulfate, colorectal cancer, gut microbiota, metabolomics, amino acid

## Abstract

Chondroitin sulfate (CS) is a well-known bioactive substance with multiple biological functions, which can be extracted from animal cartilage or bone. Sturgeon, the largest soft bone animal with ~20% cartilage content, is a great candidate for CS production. Our recent study confirmed the role of sturgeon chondroitin sulfate (SCS) in reducing colorectal cancer cell proliferation and tumor formation. Here, we further studied the effect of SCS on modulating gut microbiome structure in colorectal cancer bearing mice. In this study, the transplanted tumor mice model was constructed to demonstrate that SCS can effectively halt the growth of transplanted colorectal tumor cells. Next, we showed that SCS significantly altered the gut microbiome, such as the abundance of Lactobacillales, Gastranaerophilales, Ruminiclostridiun_5 and Ruminiclostridiun_6. According to linear discriminant analysis (LDA) and abundance map analysis of the microbial metabolic pathways, the changes in microbial abundance led to an increase of certain metabolites (e.g., Phe, Tyr, and Gly). Fecal metabolome results demonstrated that SCS can significantly reduce the amount of certain amino acids such as Phe, Pro, Ala, Tyr and Leu presented in the feces, suggesting that SCS might inhibit colorectal cancer growth by modulating the gut microbiome and altering the production of certain amino acids. Our results revealed the therapeutic potential of SCS to facilitate treatment of colorectal cancer. This study provides insights into the development of novel food-derived therapies for colorectal cancer.

## 1. Introduction

Cancer is a major problem that seriously endangers human health [1]. Among them, colorectal cancer (CRC) is a common gastrointestinal cancer, with increasing rates of morbidity and mortality worldwide in recent years. It is characterized by rapid progress, a high mortality rate, and extremely poor prognosis [2]. Clinically, many therapeutic strategies, such as antibiotics, radiotherapy, chemotherapy, targeted therapy, and immunotherapy are available, but controlling tumor growth and metastasis remains challenging, and toxic side effects of the drugs also need to be addressed [3]. Therefore, it is important to develop new approaches for CRC prevention and/or treatment. Of all the factors, gut microbiota is a key factor influencing colon tumorigenesis [4]. The regulatory role of changes in gut microbiome in colorectal cancer pathogenesis has been extensively studied. Colorectal cancer reduces microbial diversity and commensal microorganisms, and increases pathogenic microbes [5]. Therefore, it is the goal of many researchers to find effective functional foods to restore the gut microbiome in colorectal cancer patients.

Microbiota refers to trillions of microorganisms living in the digestive tract of the body. They form a relatively stable gut microecology [6]. The stability of microbiota is related to host age, genetic background, diet, and living environment. The host can affect the structure and function of microbiota, and gut flora can also affect the physiological activities of the host [7]. In recent years, more and more evidence shows that gut microenvironment is the main factor affecting the progression and treatment of malignant tumors such as colon cancer. It has been found that compared with the gut microbiota of healthy subjects, the gut microbiota of CRC patients can accelerate cell proliferation in animal models [8]. In addition, mixed feeding or fecal microbiota transplantation (FMT) strategies (containing Bifidobacterium lactis, Lactobacillus acidophilus and Staphylococcus epidermidis and gut colonizing Bifidobacterium) can affect the tumor growth of mice [9]. Antibiotics also can interfere with gut microbiota and affect the efficacy of tumor progression [10]. Therefore, we believe that the diversity and composition of gut microbiota and its metabolic function interact with tumor progression.

A growing number of studies have recognized that polysaccharides such as Astragalus Polysaccharides and microbial extracellular polysaccharides can regulate the physiological activities of mammalian host. They can enhance the system immune function and slow down the development of diseases by regulating the gut microbiota [11]. Chondroitin sulfate (CS) is the main component in cartilage tissue. It is an acidic mucopolysaccharide, which can enhance immune function and inhibit tumor growth. Kantor et al. (2016) found that CS supplementation may be directly associated with reducing the risk of CRC [12]. Zhou et al. (2021) analyzed data from the Cancer Genome Atlas (TCGA) and gene expression synthesis (GEO), and determined that CS can increase apoptosis in HCT-116 cells [13]. Our previous studies have shown that sturgeon derived chondroitin sulfate (SCS) can inhibit the tumor growth of colorectal cancer in mice [14]. However, its regulatory effect on gut microbiota largely remains unknown.

In this work, we determined the change of tumor growth in mice carrying colorectal HT-29 tumor with or without SCS supplementation. The changes of gut microbiota structure and its metabolic profile were also analyzed. The correlation between the changes of gut microbiota and fecal metabolites was evaluated.

## 2. Results and Discussion

### 2.1. Anti-Tumor Effect of SCS in HT-29 Transplanted Tumor Mice

To study the inhibitory effect of SCS on tumor growth in vivo, a tumor xenograft model was established in the present study. Based on the preliminary tests, we selected 200, 400, and 800 μg/g body weight/day as SCS treatment doses and 7 weeks as the duration of this experiment (Figure 1A). Compared with the model group, oral supplementation of SCS can significantly reduce the mortality of mice (Figure 1B), which suggests that SCS can slow down the development of colorectal cancer. The reduced expression level of tumor biomarkers CEA (Figure 1C) and CA19-9 (Figure 1D) in SCS treated mice also confirmed the anti-tumor activity of SCS. Compared to the model group, the expression of CAE decreased by ~2.1-fold in high dose group, and CA19-9 significantly decreased by ~2.7-fold. CEA and CA19-9 are important tumor biomarkers, which play an important role in the diagnosis of malignant tumors [15], monitoring the patient’s condition, and evaluating the effect of clinical treatment. Their expression level is closely related to tumor differentiation, invasion, and tumor nodule metastasis (TNM) stage [16]. Cisplatin (DDP) is one of the most active chemotherapeutic drugs available to treat a variety of malignancies [17]. CEA and CA19-9 in the high dose group also showed significant reductions compared to those of the positive control DDP group. Therefore, SCS treatment may inhibit the differentiation and invasion of tumor cells. Meanwhile, Jiang et al., (2021) reported that cucumber focused CS at 400 μg/g can improve the immune function by up-regulating the NF-κB pathway with little toxicity to the host [18]. Similarly, in our study, SCS at low dose (200 μg/g) also protect the host against tumorigenesis, suggesting that SCS is a promising candidate for cancer treatment.

In addition, the concentration of the blood SCS was measured to determine the in vivo absorbability and bioavailability of the SCS samples. As shown in Figure 1Ea, no SCS was found in the blood circulation prior to the administration of SCS. After gavage, the concentration of SCS in blood gradually increased in a time-dependent manner. Notably, only 10% of the SCS was detected in the blood. Due to the lack of enzymes to digest complex carbohydrates in humans and animals, most polysaccharides pass directly through the anterior digestive tract to the more microbial dense distal colon and rectum [19] where polysaccharides provide nutrition for the growth and metabolism of gut microbiota [20]. Similarly, the live imaging data in our results also showed that the SCS was enriched in the colon (Figure 1Eb). In addition, gut microbiome are also able to metabolize polysaccharides to generate many metabolites such as short-chain fatty acids (SCFAs), amino acids, phenolic acids, and other bioactive chemical compounds, some of which play an important role in regulating and improving the body’s physiological function and health [21]. Ding et al., (2021) found that Astragalus polysaccharides can alleviate the proliferation of melanoma by increasing fecal creatine and reducing the abundance of amino acids [11]. Combined with our results, we speculate that SCS may slow down the development of colon cancer by regulating the gut microbiome.

### 2.2. Effect of SCS on Colon Tissue in Colorectal Cancer Graft Mice

After sacrificing the mice, the distal part of their colons were removed and the depth of crypt was counted by H&E staining. Crypt depth (Figure 2A) and abnormal crypt foci (Figure 2B) were compared to the control group. Crypt depth of the model group 0.475 ± 0.088 mm was significantly higher than that of the control group 0.102 ± 0.012 mm. The SCS treatment group could reduce the colonic crypt depth. The high dose group was 0.125 ± 0.016 mm, which was restored to the level of the control group (Figure 2A). Meanwhile, severe histological lesions and inflammatory cell infiltration were observed in the model group, and SCS treatment reduced histopathological manifestations of the colon (Figure 2C). This indicates that SCS has the potential to protect colonic tissue, maintaining the barrier function of the intestinal mucosa and the structural integrity of the intestinal mucosa. Intact intestinal structure can effectively maintain the balance and diversity of the intestinal flora [22]. Therefore, our results suggest that SCS might alleviate the development of colon cancer by protecting intestinal mucosal tissue, thereby maintaining and regulating gut microbiota diversity.

### 2.3. SCS Treatment Alters the Structure of Gut Microbiota

Recent studies have indicated that a dysregulation of gut microbiota plays a critical role in the development and the progression of several diseases, including colon cancer and liver cancer [6,23]. Therefore, we sought to investigate whether SCS treatment can affect the composition of gut microbiota. We used 16S rRNA sequencing technology to study the microbiota of colonic content of mice in each group. As shown in Figure 3A, the differences in microbial composition among the four groups are clearly shown in the Venn diagram. The number of OTUs in the Control, Model group, DDP group, and the High group were 975, 817, 1033 and 1411, respectively. Model group had the lowest OUT number, indicating low microbial diversity in the Model group. However, the OTU value of DDP group is higher than that of Control group, suggesting that higher diversity does not always correlate with better disease outcome.

In order to clarify the effect of SCS on gut microbiota in tumor-bearing mice, we analyzed the community structure at different taxonomic levels. The results of gut microbiota analysis at phylum level showed that Bacteroidetes and Firmicutes were the dominant phyla in each group (Figure 3B). Compared with control group, Actinobacteria in Model group increased significantly (20.07 ± 2.07), and the relative abundance of Actinobacteria decreased significantly after SCS treatment (High group, 0.31 ± 0.06). Actinobacteria are important prokaryotes, which can synthesize metabolites, such as antibiotics, pigments, and enzyme inhibitors. They are also an important group of pathogens in mammals. They can invade the host through damaged tissue, causing diseases such as allergic pneumonia [24].

Conversely, compare to model group, SCS treatment resulted in a decreased abundance of Firmicutes (40.57 ± 4.32), whereas model group showed decreased abundance of Bacteroidetes (22.74 ± 5.90). One study showed that increased Firmicutes to Bacteroidetes ratio (F/B) was frequently observed in colon cancer or colitis patients [25]. The ratio of F/B decreased by 3.52-fold in the high group (~0.69) compared to model group (~2.43) (Figure 3B), indicating that SCS could restore the balance between Firmicutes and Bacteroidetes in tumor-bearing mice.

Our results also showed that the relative abundance of Fusobacteria in Model group was higher than that in other groups, and Fusobacteria was not found in the Control group (0.13 ± 0.06). After SCS treatment (High group), the relative abundance of Fusobacteria decreased by ~100-fold. Fusobacteria are common human pathogens, can participate in a variety of infections, such as appendicitis and inflammatory bowel disease [26,27]. Recent studies found Fusobacteria can promote colon tumor formation in animal models, and is abnormally enriched in colon cancer patients [28]. In our results, Fusobacteria was not detected in the control group, and a significant reduction of Fusobacteria was observed after treatment with SCS, consistent with what was previously reported.

At the family level (Figure 3C), compared to control group, model group showed reduced Bacteroidales_S24-7_group, Rikenellaceae, and Prevotellaceae whereas significantly increased Coriobacteriaceae, Streptoccaceae, and Bacteroidaceae. Oral administration of SCS significantly increased the abundance of Bacteroidaceas, Lachnospiraceae, and Prevotellaceae in tumor-bearing mice. These observations were in accordance with the findings reported by Mou et al. [29] regarding the structural alteration of gut microbiota by chondroitin sulfate and its oligosaccharide, fucosylated CS mainly increased the proportions of Prevotellaceae.

To demonstrate in detail the modulations of gut microbiota by SCS, we next compared the bacterial compositions at the genus level (Figure 3D). In control group, Lachnospiraceae_NK4A136_group, Lactobacillus, and Bacteroidales_S24-7_group were the predominant genus. While the Model group is mainly composed of Lachnospiraceae_NK4A136_group, Lactobacillus, Enterorhabdus, and Ruminococcaceae_UCG-014. Compared with model group, Peptococcaceae_uncultured, Lachnospiraceae_NK4A136_group, Ruminiclostridium_5, and Bacteroides in high dose group increased significantly, whereas the abundance of Ruminococcaceae_UCG-014, Ruminiclostridium_5, and Rikenellaceae_RC9_gut_group decreased to the level of control group.

S24-7 is a major Bacteroides member, which is reported to be related to abetter response to immunotherapy. By constructing a tumor bearing mouse model of lung cancer, Huang et al., (2021) found that after supplementation with ginseng polysaccharides, S24-7 in the mouse intestine increases significantly [30]. S24-7 belongs to butyrate producing bacteria, and the increase of butyrate can change the pH of the gut environment. A weak acidic environment is favorable for the growth of probiotics such as Ruminiclostridium_5. This study also found that the relative abundance of Ruminiclostridium_5 in the intestine of SCS treated colon cancer mice also increased. Ruminiclostridium is a strictly anaerobic bacterium known for its decomposition activity [31]. Although their effects on the host physiology have not been extensively studied, a prebiotic diet has found a rapid and steady increase in the relative abundance of Ruminiclostridium 5, positively correlated with the concentration of bile acids in feces [32,33]. Our current results, combined with previous research results, show that SCS can promote a healthy gut microbial ecology. Ruminococcaceae_UCG_014 is the most abundant species in the family of Erythrococcus aceae, with a wide range of carbohydrate degradation activity [34]. It may be associated with infection and inflammation and is present with high abundance in patients with inflammatory bowel disease [35]. Ruminococcaceae_UCG_014 is one of the few highly abundant bacteria (>1%) in this study, mainly enriched in the model group. These results suggest that SCS might inhibit the development of inflammation in CRC mice by enhancing the relative abundance of the Bacteroidales_S24-7_group and Ruminiclostridium_5 while inhibiting Ruminococcaceae_UCG-014. Zhou et al., (2019) found that after DDP treatment, Bacteroidales_S24-7_group was enriched and the relative abundance of Lactobacillus decreased [17]. Consistent with our results, DDP could further reduce the content of beneficial bacteria compared to the model group.

At the same time, we use the taxonomic branch map to show the taxonomic hierarchical relationship of the main taxons from phylum to genus (from inner ring to outer ring) in the sample community (Figure 4A). We further used the linear discriminant analysis (LDA) effect size (LEfSe) tool to characterize the change of the gut microbiota in colorectal cancer mice (Figure 3B). The result showed that Bacteroidales, Rikenellaceae, Enterococcus, and Butyricicoccus were enriched in control group, whereas Staphylococcus, Bacillales, Peptococcaceae, and Mollicutes were enriched in model group mice (Figure 4B). The results of LEfSe show that the key bacterial types that led to gut microflora imbalance in model group were Peptococcaceae (LDA score 3.91, *p* = 0.033) and Mollicutes (LDA score 3.79, *p* = 0.027). The genus Ruminiclostridium_5 in the high group had the highest LDA score of 4.27 (*p* = 0.025), which might be related to therapeutic effect of SCS on colorectal cancer.

Previous evidence showed that the development of colorectal cancer correlates with the imbalance of gut microflora, which has been proven to be an important factor in gut inflammation caused by colorectal cancer [36]. When gut inflammation occurs, the diversity of gut microflora will be affected, which will result in the development of colitis and colorectal cancer [37]. Pharmacological and clinical studies have consistently shown that polysaccharides have immunoregulatory activity. The gut microbiota is good at using polysaccharides as an energy source and plays a key role in disease development [38]. Therefore, the regulation of gut microbiota has become a new target to control the progression of colorectal cancer [39]. Our results demonstrate that SCS can restore the gut microbiota altered by colorectal cancer cells (Figure 4). It is speculated that SCS may alleviate colon cancer by restoring the structure of gut microbiome.

### 2.4. Effect of SCS on Gut Metabolites in Tumor-Bearing Mice

Based on the microbial abundance analysis, we performed a KEGG enrichment analysis of metabolic pathways present in those microbes whose abundance were significantly altered after SCS treatment in Figure 3B and found significant differences in the predicted cecal microbial metabolic activity between the high and model groups (Figure 4C). The supplementation of SCS was enriched for several scoring microbial pathways associated with amino acids, carbohydrates, nucleotide, and lipid biosynthesis, as well as the generation of precursor metabolites, which might contribute to the protective effect of SCS in colorectal cancer.

#### 2.4.1. Fecal Metabolic Profiling by UHPLC-Q-TOF/MS

The positive ion mode (ESI +) and negative ion mode (ESI -) ESI tests were used to test the quality control (QC) samples of each group of samples to verify the reliability of the system. The total ion chromatography (TIC) spectra of QC samples were superimposed and compared. The results show that the response strength and retention time of QCS overlap, indicating that the method has good stability and low variability caused by instrument error. Typical metabolite spectra of urine samples obtained by LC-MS are shown in Appendix A. These peaks are well separated from each other, indicating that the chromatographic and mass spectrometry conditions are suitable for the measurement of samples in this study. In the extracted-ion features, the metabolites having more than 50% missing measurement values were not used in subsequent analysis.

The overall distribution trend among all samples can be observed using principal component analysis (PCA) [40]. PCA showed that the fecal metabolic groups of mice in each group were significantly different (Figure 5A), and the samples of each group were closely clustered together. The distance between the high group and the control group was close, indicating that SCS can restore the gut metabolic profile to the level of control group. It can also be speculated that the antitumor mechanism of SCS is related to the changes of some metabolite levels.

#### 2.4.2. Detection and Identification of Differentially Produced Metabolites after SCS Treatment

Based on the P-value, the production pattern of each metabolite in influence intensity and interpretation ability was measured in order to identify differentially produced metabolites. In our study, differentially produced metabolites were selected according to the parameter’s VIP > 1 and *p*-value < 0.05 (significantly different metabolites). Several potential urine metabolites were identified by comparing the retention time (RT) and mass spectra of real standards with the standard MS/MS spectra in the database (HMDB, http://www.hmdb.ca/; MassBank, http://www.massbank.jp/).

In order to better present the differentially produced metabolites, we created a heat map (Figure 5B). Specifically, heat map analysis showed that 34 metabolites with significantly altered production were identified among control, model, positive, and high group mice.

Specifically, amino acids, such as Phe, Pro, Ala, Tyr, and Leu were also highly produced in the model group. Higher abundances of Taurohyodeoxycholic acid, Fucosyltransferase, Proscillaridin, 3-phosphonopropionic acid, and 7-deacetylkhivorin were observed in the high group. Phe and Leu, as essential amino acids, cannot be synthesized by the host and must come from either food or microbial biosynthesis. Yachida et al., (2019) explored the role of the bacterial flora in amino acid metabolism by analyzing the bacterial flora abundance [41]. Their results indicate that the gut microbes involved in phenylalanine biosynthesis were significantly increased in the gut microbiota of colon cancer patients compared to those of the healthy group. In addition, phenylalanine significantly increased in colon cancer patients, which is similar to the gut of the model group in this study. Therefore, SCS might induce structural changes in gut microbiota and decreased relative abundance of phenylalanine in fecal metabolites.

#### 2.4.3. SCS Inhibits the Production of Amino Acid Metabolites

In order to show the correlation between metabolites in more detail, we used the analysis method based on Pearson correlation to calculate the correlation coefficient between two significantly different metabolites. The correlation between two significantly different metabolites is shown in the form of correlation coefficient matrix heat map (Figure 5C). At the same time, functional prediction was performed using KEGG pathway enrichment analysis (Figure 5D), the results show that Protein digestion and absorption, Central carbon metabolism in cancer, Biosynthesis of amino acids, and Aminoacyl-tRNA biosynthesis are the main ways to affect SCS to inhibit the development of colorectal cancer.

The matrix shows the correlation between those 34 significantly different metabolites. As shown in Figure 5C, the production of N-Acetylglutamic Acid, Monolinone, and Citrazinic Acid was positively correlated with amino acids (e.g., L-Alanine, Leucine and L-Proline). On the contrary, the production for Fucosyltransferase was negatively correlated with amino acids. Therefore, our results suggest that the occurrence of colorectal cancer was accompanied by a significant change in amino acid metabolism. Among them, microbial metabolism can produce Phe, Tyr, Gly, Ala, and Ser, which can affect the host physiological function through the gut microbiome. By artificially controlling amino acid content, Muthusamy et al., (2020) found that the decrease of Ala level is a potential mechanism to inhibit the growth of MPC tumor cells [41,42]. Combined with our results, SCS may potentially alleviate colon cancer by regulating the amino acid metabolism of the gut microbiome.

Similar to anaerobic bioreactors, the large intestine has a large diversity of microbiota—more than one trillion cells—which can produce a very wide range of small molecules (i.e., metabolites) and affect many important pathways related to energy balance, nutrients uptake, and immune regulation [39]. In addition, there is increasing evidence that the microbiome and its metabolome contribute to the tumorigenesis of colorectal cancer [43]. Fecal metabolome effectively reflects the direct interaction between genetic, environmental, and dietary factors [44]. Previous studies have also documented increase in amino acids such as Phe, Pro, Leu, and Glu in CRC patients [45]. Similarly, in our results, the content of Phe, Pro, Ala, and Leu in the model group increased significantly, and decreased after SCS treatment. Although the health-promoting effect of SCS on host’s physiological function have long been described [46], no studies have investigated the effect of SCS treatment on gut microbiota. Therefore, the current study shows for the first time that SCS treatment can greatly change the composition and metabolism of mouse gut microbiota, which may contribute to the health-promoting effects of SCS.

Meanwhile, amino acids, as part of the main nutrients, can regulate the balance of energy and protein in the body. It can also be further utilized by the intestinal bacteria and the host to maintain the growth and survival of the bacterial in the gut [44]. For example, Enterorhabdus and Peptococcaceae are the most common species involved in amino acid fermentation, which can decarboxylate levodopa (L-DOPA) through the conserved tyrosine decarboxylase (TyrDC) to produce Phe in the intestine which has a negative effect. Accumulation of excess amino acids can promote disease progression by inducing inflammation [47]. In our research, similar results were found, where Phe was significantly higher in the model group than in control group and decreased after SCS treatment. Gut microbiota analysis also showed high relative abundance of Enterorhabdus and Peptococcaceae in the model group.

The above results suggest that SCS may act by changing the gut microbiota structure and reducing the production of microbial metabolites such as Phe, Pro, Ala, and Leu, regulating the carbon and amino acid metabolic pathways, which in turn inhibit tumor proliferation and development.

## 3. Materials and Methods

### 3.1. Reagents and Materials

Hybrid sturgeon (Acipenser schrenckii × Huso dauricus) was obtained from the Sturgeon Farm of Beijing Fisheries Research Institute (Fangshan, Beijing, China). Sturgeon chondroitin sulfate (SCS) was prepared according to previous published method [14,48]. All other chemical reagents were purchased from Sigma (St. Louis, MO, USA) of analytical grade. Fish CS ELISA Kit was purchased from Shanghai Enzyme-linked Biotechnology Co., Ltd. Histonstin-SP IHC kit was purchased from ZSGB-Bio (Beijing, China).

### 3.2. Animal Experiment Design

A total of 50 4-week-old Specific pathogen Free (SPF) male BALB/c nude mice (Vital River Laboratory Animal Technology, Beijing, China) were fed chow and water ad libitum. The mice were acclimated to the housing conditions (temperature of 27 ± 1 °C under a 12-h-light/-dark cycle) for a week. All experimental procedures were designed in accordance with the U.S. NIH Guidelines for the Care and Use of Laboratory Animals and the study protocol [SYXK (Jing) 2015-0046] approved by the Department of Laboratory Animal Science Ethics Committee of Peking University (Beijing, China). The experimental disease model was established via HT-29 xenograft according to our previous study [14]. Briefly, the mice were randomly divided into five treatment groups with ten mice in each group. After one week of adaptive feeding, except the control group, all mice were subcutaneously injected with HT-29 cells to induce xenograft tumors for 2 weeks. Then, different treatments were given for five weeks: (1) control group: intragastric administration of 0.1 mL/day saline; (2) disease model group: intragastric administration of 0.1 mL/day saline; (3) high dose: intragastric administration of 800 μg/g body weight/day SCS; (4) medium dose: intragastric administration of 400 μg/g body weight/day SCS; (5) low dose: intragastric administration of 200 μg/g body weight/day SCS; (6) positive control DDP group: intragastric administration of DDP. The body weight, food intake, and tumor size were measured every week, and the mice morbidity and mortality were monitored daily during the experimental period. In the end of the experiment, fecal samples were collected from all mice and immediately stored at −80 °C for subsequent analysis.

The concentration of SCS in blood at different time was measured by Fish CS ELISA Kit (Shanghai Enzyme-linked Biotechnology Co., Ltd., Shanghai, China) according to the manufacturer’s instruction.

Immunohistochemical (IHC) analysis was performed according to our previous study [14]. Briefly, after rinsed with PBS, mice colon tissues were fixed in 4% paraformaldehyde for 24 h and then rinsed with water for 6 h. Subsequently, after dehydrating with different concentration of alcohols, colon tissues were embedded in paraffin. Embedded tissues were sectioned for 5 μm and stained with hematoxylin and eosin (H&E).

### 3.3. Gut Microbiota Analysis

The TIANamp stool DNA kit (Tiangen Biotech Co. Ltd., Beijing, China) was used to extract total genomic DNA from collected caecal content and the primers 338F 5′-ACTCCTACGGGAGGCAGCAG-3′ and 806R 5′- CCGTCAATTCMTTTRAGTTT-3′ were used to amplify the V3-V4 hypervariable region in bacterial 16S rRNA genes. Equal-density-ratio mixtures of the PCR products were used to create sequencing libraries, to which index codes were assigned. The library was sequenced using Illumina Miseq platform to generate 200–450 bp paired-end reads (Bioprofile Co. Ltd., Shanghai, China). After removing the chimeric and barcode sequences, the effective tags were obtained. The QIIME2 software was used to analyze the sequences. Sequences displaying more than 97% similarity were allocated to the same operational taxonomic unit (OTU) and a typical sequence for each OTU was chosen for additional characterization. The SILVA Database was used to retrieve taxonomic data and the complexity of species diversity was calculated using the MOTHUR software according to the following indices of alpha diversity.

The KEGG analysis was performed using the software PICRUSt2 (Phylogenetic Investigation of Communities by Re-construction of Unobserved States), specifically to align the 16S gene sequence, construct the evolutionary tree, and infer the gene function spectrum of their common ancestor. The results were obtained after annotation on the KEGG da-tabase, and the abundance of secondary functional pathways was analyzed.

### 3.4. LC-MS/MS Analysis (HLIC/MS)

Fecal metabolomics profiling was analyzed using a UPLC-Q-TOF MS system (UHPLC, 1290 Infinity LC, Agilent Technologies, Santa Clara, CA, USA) coupled to a quadrupole time-of-flight (AB SCIEX Triple TOF 5600, Framingham, MA, USA), and hydrophilic interaction liquid chromatography (HILIC). Samples were analyzed using a 2.1 mm × 100 mm ACQUIY UPLC BEH 1.7 μm column (Waters, Ireland). The flow rate was 0.5 mL/min and the mobile phase contained: A = 25 mM ammonium acetate and 25 mM ammonium hydroxide in water and B = acetonitrile. The gradient was 95% B for 0.5 min and was linearly reduced to 65% in 7 min, and then reduced to 40% and maintained for 1 min, and then increased to 95% in 1.1 min, with 5 min re-equilibration period employed.

Both electrospray ionization (ESI) positive-mode and negative mode were applied for MS data acquisition. The ESI source conditions were set as follows: Ion Source Gas 1 and 2 were 60 psi, curtain gas was 30 psi, the source temperature was 600 °C, IonSpray Voltage Floating ± 5500 V. In MS only acquisition, the instrument was set to acquire over the m/z range 60–1200 Da, and the accumulation time for TOF MS scanning was set at 0.15 s/spectra. In auto MS/MS acquisition, the instrument was set to acquire over the m/z range 25–1200 Da, and the accumulation time for production scan was set at 0.03 s/spectra. The production scan was acquired using information dependent acquisition with high sensitivity mode. The Collisional Energy was fixed at 30 eV, Declustering potential was set as ± 60 V, exclude isotopes within 4 Da, Candidate ions to monitor per cycle was 6. The raw MS data were converted to MzXML files using ProteoWizard MSConvert and processed using XCMS for feature detection, retention time correction, and alignment. The metabolites were identified by accuracy mass (<25 ppm) and MS/MS data which were matched with a standards database (HMDB, http://www.hmdb.ca/; MassBank, http://www.massbank.jp/).

### 3.5. Differential Metabolites Identification and Pathway Analysis

In the extracted-ion features, only the variables having more than 50% of the nonzero measurement values in at least one group were kept. The collected GC/MS data were imported into SIMCA-P 14.1 software for principal component analysis (PCA), which was the least square-Discriminant analysis (PLS-DA) and orthogonal partial least squares discriminant analysis (OPLS-DA). Using the characteristic variable projection importance (VIP) value of OPLS-DA model > 1 and the P value of two tailed student’s *t*-test < 0.05 to find the differentially expressed metabolites.

To identify the perturbed biological pathways, the differential metabolite data were performed pathway enrichment analysis using KEGG database (http://www.genome.jp/kegg/). KEGG enrichment analyses were carried out with the Fisher’s exact test, and FDR correction for multiple testing was also performed. KEGG online database is used for pathway enrichment analysis to explore the most relevant paths and potential mechanisms. Fisher’s exact method is used to calculate the signal importance of enrichment pathway. Path enrichment analysis determines all matching paths according to the *p* Value and path enrichment factor in path topology analysis. Generally speaking, the smaller the *p* Value is, the more significant the enrichment degree is. The size of the enrichment factor indicates the reliability of significance.

### 3.6. Statistical Analysis

Data were processed using PRISM 6 statistical software (GraphPad Software, San Diego, CA, USA) and expressed as mean ± SEM (standard error of mean). For comparisons between the average values of different groups, one-way analysis of variance (ANOVA) with Duncan’s multiple range test was applied. In all analysis, the *p* < 0.05 was considered to have a significant difference.

## 4. Conclusions

In this study, a colorectal cancer bearing mice model was established and in vivo results showed that SCS supplementation decreased the proliferation of colorectal cancer. Further analysis of gut microbiota diversity and metabolite profiles were determined to evaluate additional health-promoting effects of SCS on tumor-bearing mice. SCS supplementation restored the gut microbiota in tumor-bearing mice to a level similar to that in control mice. In addition, SCS supplementation also significantly changed the metabolites profiles of large intestine in tumor-bearing mice. Our data suggests that SCS might exhibit its anti-tumor activity by remodeling the composition and the metabolic capacity of gut microbiota. In future research, more in-depth studies should be developed to provide detailed understanding of the relationship among SCS, gut microorganisms/metabolites and cancer treatment. Also, studies that specifically focus on clarifying the key microorganisms, which direct response to SCS and regulate tumor symptoms, should be established.

## Figures and Tables

**Figure 1 ijms-23-03723-f001:**
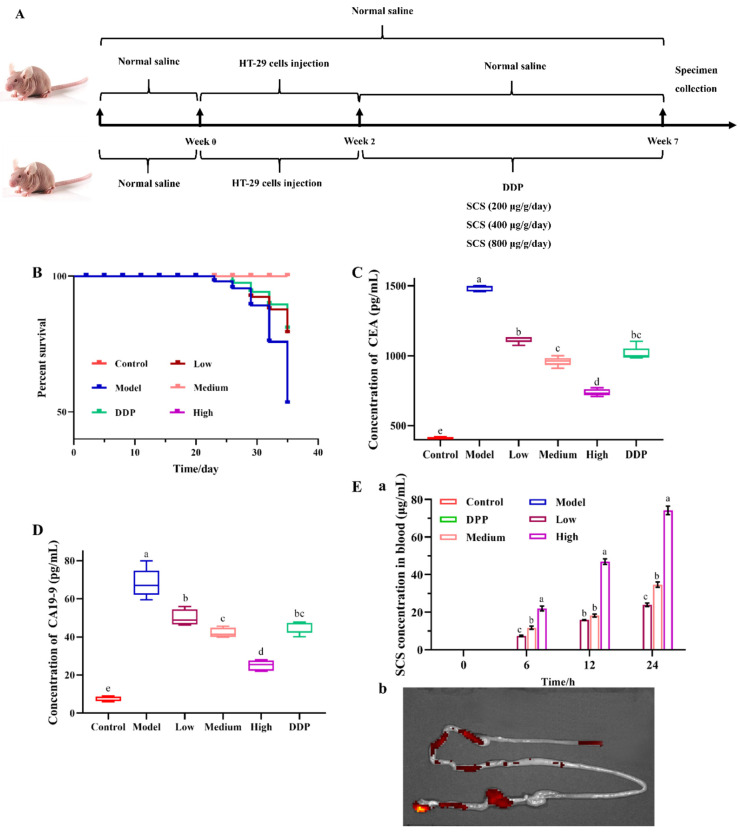
Effect of SCS on HT-29 xenograft tumor mice. Colon cancer HT-29 cells were injected into BALB/c nude mice to develop the CRC model, while the control group was injected with normal saline. After 3 weeks, mice received an intragastric administration of different doses of SCS (spine) and cisplatin (DDP) as a positive control for 4 weeks. Diagram of the experimental design (**A**), survival curve (**B**), expression of tumor biomarkers of CEA(**C**) and CA19-9(**D**), the content of SCS in blood after oral administration of SCS samples (**Ea**), ex vivo fluorescence images of major organs and tumor dissected from mice at 24 h (**Eb**). Results are expressed as mean ± SEM (*n* = 5). Different superscript letters for each column indicate significant differences (*p* < 0.05) as compared to every other group.

**Figure 2 ijms-23-03723-f002:**
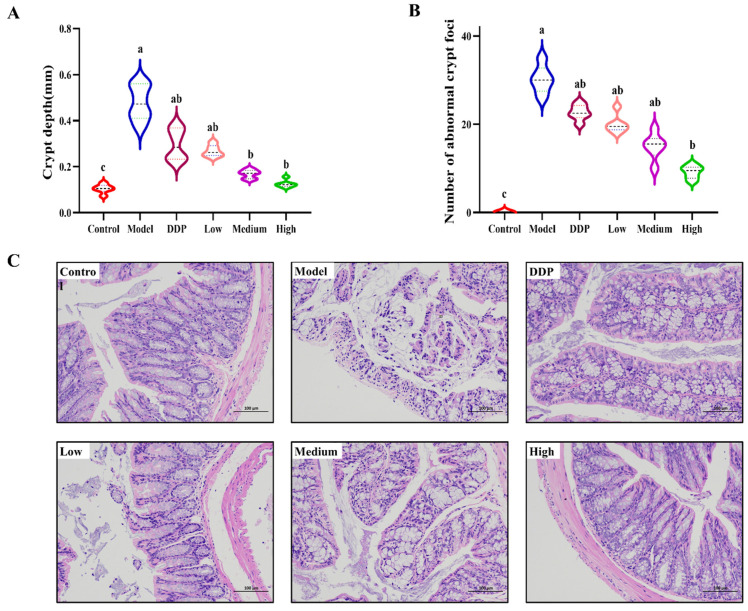
Effect of SCS on the colorectum in a mouse model of HT-29 allograft tumors. Effects of SCS treatment on the crypt depth in colorectal epithelial tissues (**A**); Effects of SCS treatment on the number of abnormal crypt foci in colorectal epithelial tissues (**B**); Typical diagrams of HE stained cells in colorectal epithelial tissues with SCS (×100) (**C**). Results are expressed as mean ± SEM (*n* = 5). Different superscript letters for each column indicate significant differences (*p* < 0.05) as compared to every other group.

**Figure 3 ijms-23-03723-f003:**
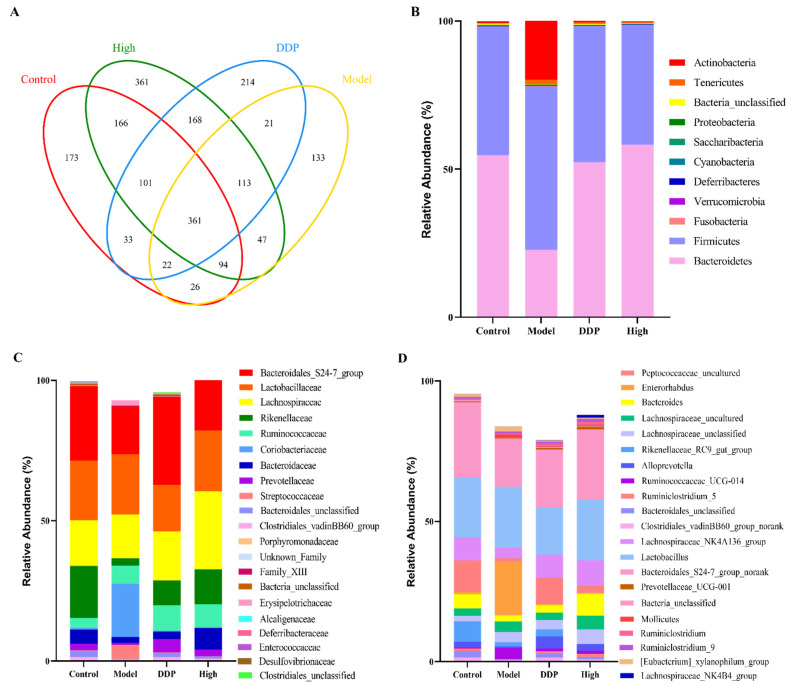
Analysis of microbiome of colon cancer mice after SCS treatment. Venn diagram illustrating overlap of OTUs (**A**); Microbiota composition at phylum level (**B**), family level (**C**), and genus level (**D**).

**Figure 4 ijms-23-03723-f004:**
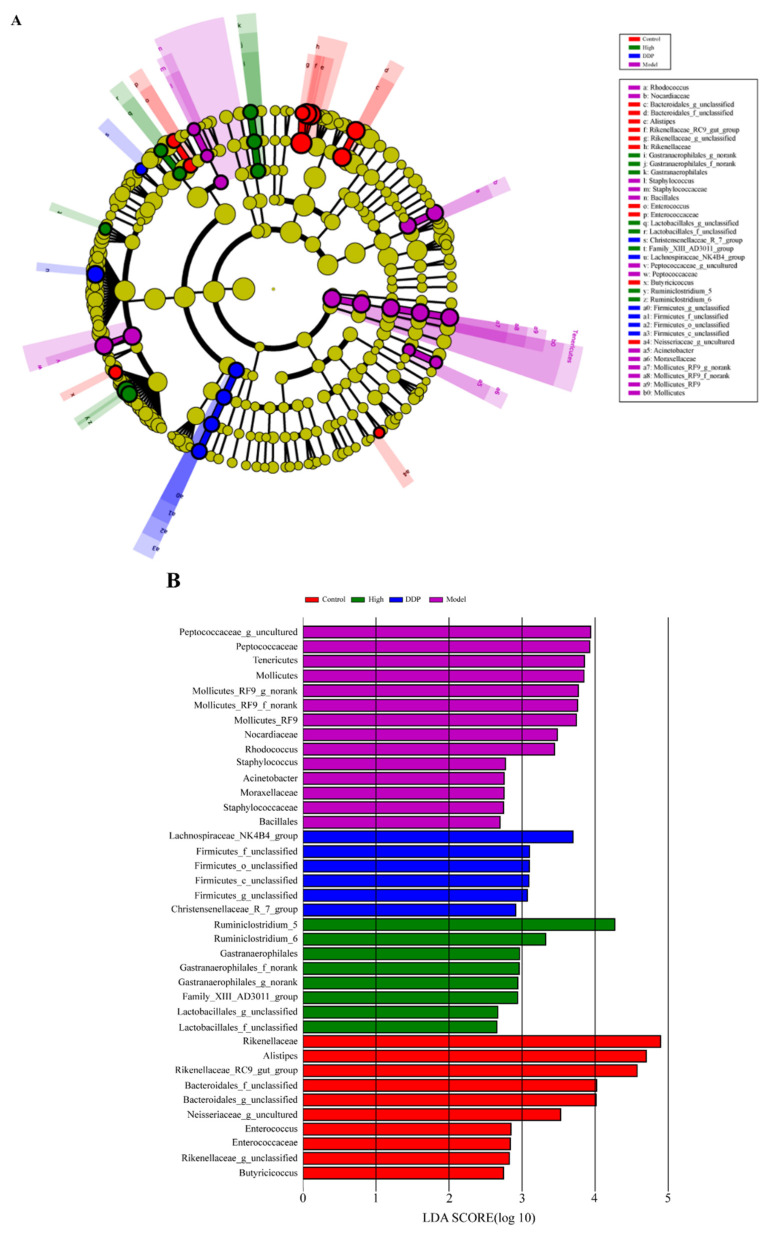
SCS regulates the composition of gut microbiota in colorectal cancer mice. Relative abundance map and difference analysis of the gut microbial composition at the genus level (LDA effect size > 2) ((**A**); Differentially abundant taxonomic clades with a linear discriminant analysis (LDA) score > 2.0 among cases with a *p* value < 0.05 (**B**); KEGG function annotation forecast (**C**).

**Figure 5 ijms-23-03723-f005:**
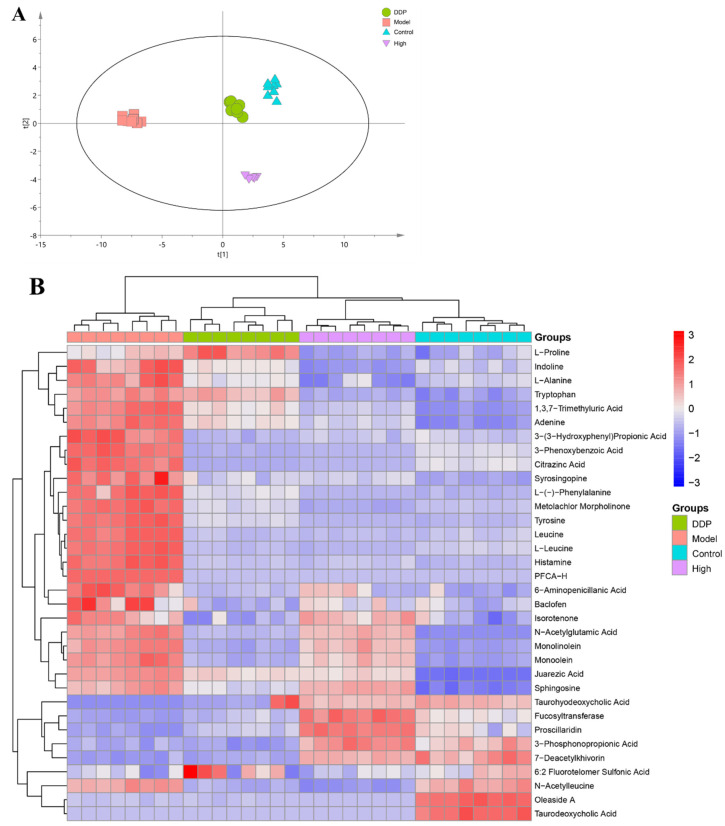
Effect of SCS on gut metabolites in tumor-bearing mice. Principal component analysis (PCA) was used to evaluate the differences of gut metabolite composition in each group (**A**). Heat map of significantly altered fecal metabolites (**B**), and the Pearson correlation coefficient value R is between −1 and +1. The correlation coefficient R between metabolites is expressed in color and circle, where R > 0 indicates positive correlation and red indicates positive correlation. R < 0 indicates negative correlation, expressed in blue; The larger the circle and the darker the color, the stronger the correlation. Correlation Clustering matrix of significant difference metabolites in each group (**C**). Enriched KEGG pathway analysis for High group vs. Model group (**D**).

## Data Availability

Not applicable.

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
