# Peer review of "Sturgeon Chondroitin Sulfate Restores the Balance of Gut Microbiota in Colorectal Cancer Bearing Mice"

_ijms, 2022, doi:10.3390/ijms23073723_

Round 1

Reviewer 1 Report

The authors have submitted an interesting work but I have some critical concerns that need to be addressed:

  1. The amount of SCS given to mice sounds very high since 800 ug/g/day means that at the end of the 7 weeks planned the treated mice would have received a total amount of around 4% of their body weight in SCS. The authors should explain their preliminary tests and justify whether these amounts of SCS could have a translational impact in humans.
  2. The role of CS for prevention of CRC has been previously reported and the doses used in others works could be compared and discussed
  3. Statistical analyses in figures 1 and 2 are missing
  4. A dose-escalating study would establish the minimal concentration able to induce the reported effects here

Minor:

  1. The introduction should be enriched with literature data about chondroitin sulphate and colorectal cancer
  2. Limitations of this work have to be clearly stated in the Conclusions section

Reviewer 2 Report

In this study, the authors have used a mouse model of colorectal cancer to investigate the anti-tumor effects of chondroitin sulfate from sturgeon. The paper submitted is the latest in a series of studies of the effects of sturgeon chondroitin sulfate in the gut. They show evidence in treated mice for increased survival, decreased cancer markers CA19-9 and CEA,  altered microbiome profiles, and altered gut metabolite profiles. These changes are proposed as potential mechanisms for the anti-tumor action of sturgeon chondroitin sulfate.

Unfortunately, the standard of presentation of this manuscript is not high enough for publication.

Abstract line 27: Please alter ‘potential of SCS to facilitate colorectal cancer’ to ‘potential of SCS to facilitate treatment of colorectal cancer’

Line 75: Section 2:1 is titled ”Anti-diabetic effects of SCS inHT-29 transplanted tumor mice.”  But the text in this section concerns anti-tumor effects, not anti-diabetic effects.

Line 109: Figure 1 is titled “Effects of SCS on body weight and food intake in HT-29 xenograft tumor mice model” but the figure does not show these results; instead it shows evidence for anti-tumor activity.  

Figures and text throughout the paper: The abbreviations ‘DPP’ and ‘DDP’ are used interchangeably and are nowhere defined. There is no discussion of the significance of DPP and DDP treated groups of mice. Is this dipeptidyl peptidase? What is its purpose and how does it work?

Figure 4c: It is not clear how the KEGG analysis in this figure was obtained; please add explanatory text to section 3.3 on Microbiota analysis.  

References:

Reference 1: In addition to the Erratum cited, please add the original 2018 paper by Bray et al.

In addition, some of the references have been checked and found to be incorrect. This is worrying and makes the manuscript very difficult to evaluate. Not all of the references were checked. The authors need to find out what has gone wrong here and supply a correct reference list. If these errors have been missed there may be many others.

Reference 4: cited on line 42 is incorrect and does not mention the gut or microbiota

Reference 8 cited on line 51 is incorrect and does not mention the gut flora or microbiome

Reference 12 cited on line 59 is incorrect and does not mention antibiotics

Reference 22 cited on lines 175 and 177 is incorrect and contains no mention of fusobacteria

Reference 38 cited on line 268 is incorrect and contains no mention of PCA

Round 2

Reviewer 1 Report

The manuscript has been revised and improved and my suggestions properly addressed.

Reviewer 2 Report

The authors have made the revisions and corrections requested.